# Efficient and Transferable Adversarial Examples
# from Bayesian Neural Networks

**Martin Gubri**[1]     **Maxime Cordy**[1]     **Mike Papadakis**[1]     **Yves Le Traon**[1]     **Koushik Sen**[2]

[1]University of Luxembourg, Luxembourg, LU
[2]University of California, Berkeley, CA, USA

## Abstract

An established way to improve the transferability of black-box evasion attacks is to craft the adversarial examples on an ensemble-based surrogate to increase diversity. We argue that transferability is fundamentally related to uncertainty. Based on a state-of-the-art Bayesian Deep Learning technique, we propose a new method to efficiently build a surrogate by sampling approximately from the posterior distribution of neural network weights, which represents the belief about the value of each parameter. Our extensive experiments on ImageNet, CIFAR-10 and MNIST show that our approach improves the success rates of four state-of-the-art attacks significantly (up to 83.2 percentage points), in both intra-architecture and inter-architecture transferability. On ImageNet, our approach can reach 94% of success rate while reducing training computations from 11.6 to 2.4 exaflops, compared to an ensemble of independently trained DNNs. Our vanilla surrogate achieves 87.5% of the time higher transferability than three test-time techniques designed for this purpose. Our work demonstrates that the way to train a surrogate has been overlooked, although it is an important element of transfer-based attacks. We are, therefore, the first to review the effectiveness of several training methods in increasing transferability. We provide new directions to better understand the transferability phenomenon and offer a simple but strong baseline for future work.

## 1 INTRODUCTION

Deep Neural Networks (DNNs) have caught a lot of attention in recent years thanks to their capability to solve efficiently various tasks, especially in computer vision [Dar-

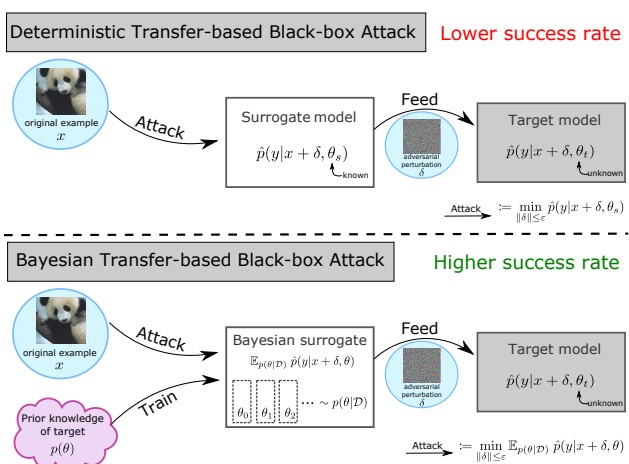

Figure 1: Illustration of the proposed approach.

gan et al., 2019]. However, a common pitfall of these models is that they are vulnerable to adversarial examples, i.e., misclassified examples that result from slightly altering a well-classified example at test time [Biggio et al., 2013, Szegedy et al., 2013]. This constitutes a critical security threat, as a malicious third party may exploit this property to enforce some desired outcome.

Such *adversarial attacks* have been primarily designed in white-box settings, where the attacker is assumed to have complete knowledge of the target DNN (including its weights). While studying such worst-case scenarios is essential for proper security assessment, in practice the attacker should have limited knowledge of the target model. In such a case, the adversarial attack is applied to a surrogate model, with the hope that the crafted adversarial examples *transfer to* (i.e., are also misclassified by) the target DNN.

Achieving transferability remains challenging, though. This is because adversarial attacks were designed to optimize the loss function of a specific model [Goodfellow et al., 2014, Kurakin et al., 2019], different from that of the target model. As a result, Liu et al. [2017] improved transferability by

*Accepted for the 38th Conference on Uncertainty in Artificial Intelligence* (UAI 2022).

attacking an *ensemble* of architectures. The key intuition is that adversarial examples that fool a diverse set of models are more likely to generalize. While ensemble-based attacks typically report significantly higher success rates than their single-model counterparts, their computational cost is prohibitive due to the necessity to independently train several surrogate models (to form a diverse ensemble).

In this paper, we analyse the unknown target model with a probabilistic eye, and relate transfer-based attacks to uncertainty. We propose a new method to improve the transferability of adversarial examples using approximate Bayesian inference to build a surrogate – and do so with less computation overhead compared to ensemble-based methods. Our approach, shown in Figure 1, leans upon recent results in Bayesian Deep Learning. More precisely, we train our surrogate with a cyclical variant of Stochastic Gradient Markov Chain Monte Carlo (i.e., *cSGLD* [Zhang et al., 2020]) to sample from the posterior distribution of neural network weights. We then perform efficient approximate Bayesian model averaging during the attack with minimal modifications of the attack algorithms.

We evaluate our approach on the ImageNet, the CIFAR-10 and the MNIST datasets with a variety of DNN architectures, four adversarial attacks, and three test-time transformations. Overall, our results indicate that applying cSGLD significantly improves the success rate compared to training single DNNs and outperforms classical ensemble-based attacks in terms of computation cost. Deep Ensemble requires at least 2.51 times more flops to achieve the same success rates as cSGLD when the targeted architecture is known. This can represent, on ImageNet, a saving of 3.56 exaflops (2.36 vs 5.92). At constant computation costs, our method increases the intra-architecture transfer success rates between 1.6 and 82.0 percentage points and the inter-architecture transfer success rates between -2.3 and 83.2. cSGLD always raises the effectiveness of test-time techniques designed for transferability between 3.8 and 56.2 percentage points. Applied alone, it is more effective than these techniques applied to a single DNN in 105/120 cases.

To summarize, our contributions are:

- We relate uncertainty and transferability of adversarial examples with a Bayesian perspective. The posterior distribution represents a belief about the unknown target model.

- We propose the first method based on a Bayesian Deep Learning technique to generate transferable adversarial examples. Existing iterative attacks can be easily modified to perform approximate Bayesian model averaging at no additional computational cost.

- We pave the way for improving surrogates at train-time by evaluating six Bayesian and ensemble techniques. cSGLD is a strong competitor, though other techniques open promising avenues.

- We advocate the use of a new metric, T-DEE, to compare the effectiveness of transferability techniques with the strong baseline of Deep Ensemble.

- Our evaluation on ImageNet, CIFAR-10 and MNIST reveals significant improvements over the single-DNN and Deep Ensemble baselines in diverse experimental settings. Our train-time method improves existing test-time techniques, and is better in most cases on a competitive basis. We open new ways to understand transferability.

## 2 BACKGROUND AND RELATED WORK

**Adversarial attacks.** We consider 4 gradient-based attacks, which aims to maximise the prediction loss $L(x, y, \theta)$ with a $p$-norm constraint: $\arg\max_{\|\delta\|_p \leq \varepsilon} L(x + \delta, y, \theta)$. FGSM [Goodfellow et al., 2014] is a L$\infty$ single-step attack defined by $\delta_{\text{FGSM}} = \varepsilon \, \text{sign}(\nabla_x L(x, y, \theta))$. Its L2 equivalent is: $\delta_{\text{FGM}} = \varepsilon \frac{\nabla_x L(x,y,\theta)}{\|\nabla_x L(x,y,\theta)\|_2}$. The adversarial example is then clipped in $[0, 1]$. I-FGSM [Kurakin et al., 2019] applies iteratively FGSM with a small step-size $\alpha$: $\delta_0 = 0$ and $\delta_{i+1} = \text{proj}_{B_\varepsilon}(\delta_i + \alpha \, \text{sign}(\nabla_x L(x, y, \theta)))$, where $\text{proj}_{B_\varepsilon}(\bullet)$ projects the perturbation in the L$p$ ball of radius $\varepsilon$. The L2 variant is derived similarly. MI-FGSM attack [Dong et al., 2018] adds a momentum term with decay factor $\mu$ to the previous attack: $g_{i+1} = \mu \, g_t + \frac{\nabla_x L(x,y,\theta)}{\|\nabla_x L(x,y,\theta)\|_1}$ and $\delta_{i+1} = \text{proj}_{B_\varepsilon}(\delta_i + \alpha \, \text{sign}(g_{t+1}))$. PGD [Madry et al., 2018] adds random restarts to I-FGSM and $\delta_0$ is sampled uniformly inside the ball $B_\varepsilon$. Figure 2 in Appendix A illustrates their relations.

**Ensemble surrogate.** Liu et al. [2017] show the benefit of ensembling architectures for inter-architecture transfer-based black-box attacks. Our work leans on theirs and complements it by demonstrating that attacking models sampled with cSGLD (performing Bayesian model averaging on a unique architecture) achieves better transferability at lesser computation cost.

**Input and model transformations.** Other approaches have been developed to improve transferability of adversarial examples. They transform the model or the input at test time (i.e., after training, when performing the attack). *Ghost Networks* (GN) [Li et al., 2018] use Dropout and Skip Connection Erosion to generate on-the-fly diverse sets of surrogate models from one or more base models. *Input Diversity* (DI) [Xie et al., 2019] applies random transformations (random resize followed by random padding) to the input images at each attack iteration. *Skip Gradient Method* (SGM) [Wu et al., 2020] favours the gradients from skip connections rather than residual modules through a decay factor applied to the latter during the backward pass. These techniques can naturally be combined to ours: (i) cSGLD can provide at a low computation cost a diverse set of base

models to build GN; (ii) DI applies transformations to adversarial inputs independently of the surrogate models; (iii) SGM modifies backward passes during the attack, independently of the training method. As our evaluation will reveal, our train-time method further improves the transferability of the above three techniques and outperforms them 87.5% of the time. It is also compatible with other test-time approaches not considered in this paper, such as linear backpropagation [Guo et al., 2020], intermediate level attack [Huang et al., 2019], Nesterov accelerated gradient and scale invariance [Lin et al., 2020], and serial mini-batch ensemble attack [Che, 2020].

**Bayesian Neural Network (BNN) and adversarial examples.** Though not our goal, past research aimed at generating adversarial examples for BNNs (we rather use Bayesian Deep Learning as a way to attack classical DNNs). Grosse et al. [2018] show that BNN uncertainty measures are vulnerable to high-confidence-low-uncertainty adversarial examples crafted on Gaussian Processes. Palacci and Hess [2018] show that several SG-MCMC sampling schemes are not secure against white-box attacks. Wang et al. [2018] use SGLD and Generative Adversarial Network to detect adversarial examples instead of crafting them.

Carbone et al. [2020] claim that BNNs are robust against gradient-based attacks because gradients vanish in expectation under the true posterior distribution. Their conclusions hold theoretically under the restrictive assumption of the large-data overparametrized limit, and experimentally for HMC and VI on MNIST and Fashion MNIST. In Appendix C, our experiments reveal opposite conclusions about cSGLD: our surrogates DNNs suffer more often from vanished gradients than our cSGLD surrogates. On MNIST, we observe that 60.6-86.6% of individual gradients of HMC or VI vanish before averaging them. Therefore, the theoretical development of Carbone et al. [2020] does not seem to explain most gradient vanishing. Furthermore, VI on larger datasets (ImageNet and CIFAR-10) do not suffer from vanishing gradients.

## 3 APPROACH

**A Bayesian perspective on transferability.** Under a specified threat model, we relate uncertainty and posterior predictive distributions to transferability. We consider a classification problem with a training dataset $\mathcal{D} = \{(x_i, y_i) \sim p(x, y)\}_{i=1}^N$ and $C$ class labels. A probabilistic classifier parametrized by $\theta$ maps $x_i$ into a predictive distribution $\hat{p}(y|x_i, \theta)$. A white-box adversarial perturbation of a test example $(x, y) \sim p(x, y)$ against such classifier is defined as:

$$\delta_\theta \in \underset{\|\delta\|_p \leq \varepsilon}{\arg\min} \, \hat{p}(y|x + \delta, \theta).$$

In practice, this optimization problem is solved by replacing the predictive distribution with a loss function (see Section 2). The *transferability* phenomenon is the empirical observation that an adversarial example for one model is likely to be adversarial for another one [Goodfellow et al., 2014]. Black-box attacks can leverage this property by crafting adversarial examples using white-box attacks against a surrogate model to target an unseen model [Papernot et al., 2016].

**Assumption 1** (Threat model). We define our threat model with the following assumptions on the targeted classifier:

1. Its architecture is known and so is its prediction function $\hat{p}(y|x, \bullet)$ [1].

2. Its training set $\mathcal{D}$ is known.

3. Its parameters $\theta_t$, estimated by maximum likelihood, are unknown.

4. A reasonable prior on its parameters $p(\theta_t)$ is known[2].

5. No oracle access (test-time feedback) is possible.

Assuming the threat model described in Assumption 1, *uncertainty on target parameters arises from the stochastic nature of training*, and more specifically from two sources of randomness: (i) every Stochastic Gradient Descent (SGD) update depends on a random batch of training examples[3], (ii) weights are randomly initialized at the beginning of training[4]. From the attacker subjective view, the target parameters obtained at the end of training are random variables.

We argue that $\theta_t$ is approximately distributed according to the posterior distribution $p(\theta|\mathcal{D})$. Mingard et al. [2020] observes a strong correlation between the probability to obtain with SGD or its variants a function consistent with a training set and the Bayesian posterior probability of this function. Mandt et al. [2017] shows that SGD with constant learning rate has a stationary distribution centred on an optimum, which approximates a posterior. Marginalizing over local optima, we obtain a posterior that is the distribution of SGD endpoints with a step decay learning rate schedule (as widely used).

Then, *the best transferable adversarial example approximately minimizes the Bayesian posterior predictive distribution* $p(y|x, \mathcal{D}) = \mathbb{E}_{p(\theta|\mathcal{D})} \, \hat{p}(y|x, \theta)$ and our black-box attack objective is:

$$\delta^* \in \underset{\|\delta\|_p \leq \varepsilon}{\arg\min} \, \mathbb{E}_{\theta_t \sim p(\theta|\mathcal{D})} \, \hat{p}(y|x + \delta, \theta_t). \tag{1}$$

---

[1] We discuss the unknown architecture case further on.

[2] In practice, it corresponds to knowing the weight decay hyperparameter, see discussion below.

[3] The same argument holds for SGD variants.

[4] Despite being independent and identically distributed random variables, weights initialization values play an important role in guiding the SGD trajectory [Frankle and Carbin, 2019].

Usually in adversarial machine learning, transferable adversarial examples are optimized against one surrogate model. This is similar to solving problem (1) deterministically by approximating the expectation of the posterior predictive with a "plug-in" estimation of the parameters, $\hat{\theta}_{\text{MAP}}$ the maximum a posteriori probability (MAP) estimate: $\delta^* \approx \delta_{\hat{\theta}_{\text{MAP}}}$. To avoid overfitting to the surrogate model, random transformations of inputs or prediction functions were developed in the literature (see Section 2).

A fundamental issue is that the closed form of the posterior predictive distribution is intractable for DNNs. Our contribution lies in *sampling from the posterior distribution to build a surrogate in black-box adversarial attacks*. We replace the crude MAP approximation of the posterior predictive distribution with a more accurate one to generate transferable adversarial examples. Therefore, we focus on the training phase by considering the methods and the computational costs of obtaining the surrogate model, whereas most previous work searches to optimize adversarial examples crafting at the time of the attack ("test-time").

**SG-MCMC & cSGLD.** In practice, we perform Bayesian model averaging using samples obtained from Stochastic Gradient-Markov Chain Monte Carlo (SG-MCMC). SG-MCMC is a family of approximate Bayesian inference techniques, inaugurated by SGLD [Welling and Teh, 2011], that combines SGD with MCMC. Adding noise during training allows to sample from the posterior distribution of parameters. The empirical distribution of the samples approximates the posterior. Then, our method aims to solve the following optimization problem:

$$\delta_{\{\theta_s\}} \in \arg\min_{\|\delta\|_p \leq \varepsilon} \frac{1}{S} \sum_{s=1}^{S} \hat{p}(y|x+\delta, \theta_s) , \qquad (2)$$

where $\{\theta_s \sim p(\theta|\mathcal{D})\}_{s=1}^{S}$ are samples of the posterior.

We choose to apply the recently proposed *cyclical Stochastic Gradient Langevin Dynamics* (cSGLD) [Zhang et al., 2020], a state-of-the-art SG-MCMC technique. cSGLD performs warm restarts by dividing the training into cycles that all start from the initial learning rate value (cf. illustration in Appendix A). Each cycle consists of (1) an exploration stage with larger learning rates which corresponds to the burn-in period of MCMC algorithms; (2) a sampling stage that samples parameters at regular intervals and operates with smaller learning rates and added noise. Starting a new cycle with a large learning rate allows the exploration of another local maximum of the loss landscape. Contrary to most SG-MCMC methods, cSGLD has the compelling benefit of sampling from both several modes of the posterior distributions and locally inside each mode, avoiding mode collapse. Another major advantage of cSGLD is that its computation overhead compared to SGD/Adam is negligible (0.019% flops for one epoch on PreResNet110 on CIFAR-10 and 0.015% for ResNet50 on ImageNet).

**Difference with Ensembling.** Our work differs from previous research [Liu et al., 2017, Li et al., 2018, Xie et al., 2019] that relates diversity with transferability in the same way that Ensembling and Bayesian Model Averaging do [Minka Thomas P., 2002]. The latter "assumes that the true model lies within the hypothesis class of the prior, and performs soft model selection [...]. In contrast, ensembles [...] combine the models to obtain a more powerful model; ensembles can be expected to be better when the true model does not lie within the hypothesis class" [Lakshminarayanan et al., 2016]. Under Assumptions 1, the unknown target model, our true model here, lies within the hypothesis class of its prior by definition. Therefore, we argue that under these conditions, a Bayesian approach is a more natural way to select a surrogate model.

**Target prior.** We express the prior of a standard target DNN. Deterministic DNNs are classically trained using the cross-entropy loss regularized by weight decay:

$$\min_{\theta_t} -\frac{1}{N} \sum_{i=1}^{N} \log \hat{p}(y_i|x_i, \theta_t) + \frac{\lambda}{2}\|\theta_t\|^2,$$

with $\lambda$ its weight decay hyperparameter. This maximum likelihood estimation (MLE) procedure corresponds to the maximum a posteriori inference (MAP) of this implied probabilistic model:

$$p(y, \theta_t|x) = p(y|x, \theta_t)p(\theta_t),$$

where $p(y|x, \theta_t)$ is the likelihood function and $p(\theta_t) = \mathcal{N}(\theta_t|0, \frac{1}{N\lambda}I)$ a Gaussian prior. Therefore, in this standard setting, the hypothesis 4 reduces to knowing the weight decay hyperparameter $\lambda$.

**Extension to unknown architecture.** Let $\mathcal{A} = \{a_i\}_i$ be a countable set of candidate architectures, $p(a)$ a prior on $\mathcal{A}$, $\theta^a$ the parameters of the architecture $a$ and $\hat{p}^a(y|x, \theta^a)$ its predictive distribution. Discarding hypothesis 1 of Assumption 1 on the knowledge of the architecture, the architecture of the target $a$ becomes a random variable. We perform *Bayesian Model Comparison* to compute the posterior over models:

$$p(a|\mathcal{D}) \propto p(\mathcal{D}|a)p(a). \qquad (3)$$

We marginalize over architectures to express the complete posterior predictive distribution as the average across architectures weighted by their posterior probabilities:

$$p(y|x, \mathcal{D}) = \sum_{a \in \mathcal{A}} p(a|\mathcal{D})p(y|x, \mathcal{D}, a)$$
$$\propto \mathbb{E}_{p(a)} p(\mathcal{D}|a) \mathbb{E}_{p(\theta^a|\mathcal{D})} \hat{p}^a(y|x, \theta^a) \qquad (4)$$

If $\mathcal{A}$ is finite and small, we can approximate this quantity with a weighted average of one cSGLD empirical posterior predictive distribution per architecture. Otherwise, we estimate it with MCMC by sampling according to $p(a)$ a finite subset $A = \{a_i \sim p(a)\}_{i=1}^{S_A} \subset \mathcal{A}$ of architectures, where the number of architectures $S_A$ is fixed by the computational budget. We sample $S$ parameters $\{\theta_s^a\}_{s=1}^S$ for all $a \in A$. Then, our inter-architecture attack that minimizes our approximation of $p(y|x, \mathcal{D})$ becomes:

$$\delta_A \in \underset{\|\delta\|_p \leq \varepsilon}{\arg\min} \frac{1}{S_A S} \sum_{a \in A} p(\mathcal{D}|a) \sum_{s=1}^{S} \hat{p}^a(y|x + \delta, \theta_s^a) \quad (5)$$

Various methods exist to approximate model evidence [Friel and Wyse, 2012]. To simplify empirical conclusions, we assume that all architectures in $\mathcal{A}$ have approximately equal evidence. This strong assumption is reasonable here, since we select widely used architectures which are well-specified on the standard benchmark datasets evaluated. For fairness to ensemble baselines, our experiments on unknown architectures do not include the target architecture in the set $\mathcal{A}$.

**Attack algorithm.** One can approximate the solution of Equations 2 and 5 with minor modifications of existing adversarial attack algorithms, i.e. simply cycling surrogate models throughout iterations. To efficiently approximate Bayesian model averaging during iterative attacks, we compute the gradient of every iteration on a single model sample per architecture. If multiple architectures are attacked, we average their gradients (see Algorithm 1 in Appendix A). The cost of iterative attacks, measured as the number of backward passes, does not increase with the number of samples $S$.

**Clarifications.** In the following, the intra-architecture transferability represents the case of known target architecture. The mass of the prior concentrates on a single architecture, thus the posterior too. Respectively, the inter-architecture transferability corresponds to an unknown target architecture not sampled in the surrogate set. The prior of the target architecture may not be zero, given the extension to unknown architecture described above. But we hold-out this architecture from the surrogate set during empirical evaluation for fairness to baseline and to simplify result interpretations.

## 4 EXPERIMENTS

The goal of our approach is to increase the transferability of adversarial examples by using a surrogate sampled from the posterior distribution to attack a deterministic DNN.

**Setup summary.** The target models are deterministic DNNs and are never used as a surrogate. For a fair comparison between DNNs and cSGLD, we train the surrogate DNNs

on CIFAR-10 and MNIST using the same process as the target models. ImageNet targets are third-party pretrained models. Each cSGLD cycle lasts 50 epochs and samples 5 models on CIFAR-10, 10 epochs/4 models on MNIST, 45 epochs/3 models on ImageNet. We report the success rate (misclassification rate of untargeted adversarial examples) averaged over three attack runs. We craft adversarial examples from correctly predicted test examples (all examples for CIFAR-10 and MNIST, and a random subset of 5000 examples for ImageNet). The iterative attacks (I-FGSM, MI-FGSM, and PGD) perform 50 iterations such that the transferability rates plateaus (Appendix G). Each attack computes the gradient of one model per architecture. Therefore, their computation cost and volatile memory are not multiplied by the size of the surrogate, except for FGSM which computes its unique gradient against all available models. The source code is publicly available[5]. Appendix A presents the experimental setup in details.

### 4.1 INTRA-ARCHITECTURE TRANSFERABILITY

Since SG-MCMC methods sample the weights of a given architecture, we expect our approach to work particularly well in settings where the architecture of the target model is known, but not its weights. To demonstrate this, we compare the intra-architecture transfer success rates of cSGLD with the ones of Deep Ensemble surrogates (using 1 up to 15 independently trained DNNs). Architectures are ResNet-50 (ImageNet), PreResNet110 (CIFAR-10) and fully connected 1200-1200 (MNIST).

Appendix D provides the detailed results for four classical gradient-based attacks on the three datasets. In summary, for a similar computation cost on ImageNet and CIFAR-10, cSGLD systematically increases the success rate of iterative attacks by 13.8 (ImageNet, MI-FGSM, L∞) to 49.2 (CIFAR-10, I-FGSM, L2) percentage points, and of FGSM by 12.18 to 22.2. On MNIST, it ranges from 6.8 to 80.5. One explanation for the highest improvements is that DNN-based L2 norm attacks suffer from vanishing gradients on CIFAR-10 and MNIST, whereas cSGLD avoids it thanks to fast convergence and warm restarts (cf. Appendix C for proportions of vanished gradients).

Inspired by DEE [Ashukha et al., 2020], we propose the **Transferability-Deep Ensemble Equivalent (T-DEE)** metric as the number of independently trained DNNs needed to achieve the same success rate as the technique considered (computed with linear interpolation). Under some assumptions[6], Deep Ensemble samples exactly from the distri-

---

[5] https://github.com/Framartin/transferable-bnn-adv-ex

[6] Besides Assumptions 1, we suppose that Deep Ensemble uses the target optimizer, and that the minimum in Eq. 2 is reached, i.e., that the attack doesn't fail due to vanished or obfuscated gradients [Athalye et al., 2018].

Table 1: Number of DNNs (T-DEE) and training computation budget (in flops) to achieve the intra-architecture transferability of cSGLD with Deep Ensemble. Higher is better. ">15" means that 15 DNNs always transfer less than cSGLD.

| Dataset | Attack | Norm | T-DEE | Flops Ratio |
|---------|--------|------|-------|-------------|
| ImageNet | I-FGSM | L2 | $4.91$ $_{\pm 0.11}$ | $2.84$ $_{\pm 0.06}$ |
|  |  | L$\infty$ | $4.34$ $_{\pm 0.13}$ | $2.51$ $_{\pm 0.08}$ |
|  | MI-FGSM | L2 | $4.69$ $_{\pm 0.18}$ | $2.71$ $_{\pm 0.10}$ |
|  |  | L$\infty$ | $4.38$ $_{\pm 0.03}$ | $2.53$ $_{\pm 0.02}$ |
|  | PGD | L2 | $5.00$ $_{\pm 0.11}$ | $2.89$ $_{\pm 0.06}$ |
|  |  | L$\infty$ | $4.42$ $_{\pm 0.16}$ | $2.56$ $_{\pm 0.09}$ |
|  | FGSM | L2 | $5.81$ $_{\pm 0.34}$ | $3.35$ $_{\pm 0.19}$ |
|  |  | L$\infty$ | $5.98$ $_{\pm 0.03}$ | $3.46$ $_{\pm 0.02}$ |
| CIFAR10 | I-FGSM | L2 | $>15$ $_{\pm nan}$ | $>15$ $_{\pm nan}$ |
|  |  | L$\infty$ | $3.76$ $_{\pm 0.08}$ | $3.76$ $_{\pm 0.08}$ |
|  | MI-FGSM | L2 | $5.56$ $_{\pm 0.80}$ | $5.56$ $_{\pm 0.80}$ |
|  |  | L$\infty$ | $2.88$ $_{\pm 0.03}$ | $2.87$ $_{\pm 0.03}$ |
|  | PGD | L2 | $>15$ $_{\pm nan}$ | $>15$ $_{\pm nan}$ |
|  |  | L$\infty$ | $3.74$ $_{\pm 0.12}$ | $3.74$ $_{\pm 0.12}$ |
|  | FGSM | L2 | $>15$ $_{\pm nan}$ | $>15$ $_{\pm nan}$ |
|  |  | L$\infty$ | $8.72$ $_{\pm 0.01}$ | $8.72$ $_{\pm 0.01}$ |
| MNIST | I-FGSM | L2 | $>15$ $_{\pm nan}$ | $>15$ $_{\pm nan}$ |
|  |  | L$\infty$ | $3.42$ $_{\pm 0.17}$ | $3.42$ $_{\pm 0.17}$ |
|  | MI-FGSM | L2 | $>15$ $_{\pm nan}$ | $>15$ $_{\pm nan}$ |
|  |  | L$\infty$ | $2.79$ $_{\pm 0.07}$ | $2.79$ $_{\pm 0.07}$ |
|  | PGD | L2 | $>15$ $_{\pm nan}$ | $>15$ $_{\pm nan}$ |
|  |  | L$\infty$ | $3.26$ $_{\pm 0.28}$ | $3.26$ $_{\pm 0.28}$ |
|  | FGSM | L2 | $>15$ $_{\pm nan}$ | $>15$ $_{\pm nan}$ |
|  |  | L$\infty$ | $>15$ $_{\pm nan}$ | $>15$ $_{\pm nan}$ |

bution of target parameters, and is thus optimal for intra-architecture transferability with infinite computing power.

Table 1 reports the T-DEE and the *computing ratio*, i.e., the total number of flops to train such DNNs ensemble divided by the number of flops used to trained cSGLD. This ratio represents the computational gain factor achieved by our approach[7]. In the worst case across the three datasets, an ensemble of 3 surrogate DNNs is required to beat the cSGLD surrogate, while requiring at least 2.51 times more flops during the training phase. On CIFAR-10 and MNIST and considering L2 attack specifically (MI-FGSM CIFAR-10 aside), it even outperforms the ensemble of 15 DNNs by a significant factor (up to 71.2 percentage points). On ImageNet, cSGLD achieves the same success rate as 4.38–5.98 DNNs, which corresponds to dividing the number of flops by 2.51–3.46.

Then, the uncertainty on parameter estimation captured by cSGLD is useful to discover generic adversarial directions.

---

[7]The ImageNet computing ratios don't equal to T-DEE since 1 DNN is trained for 130 epochs and cSGLD for 225.

## 4.2 INTER-ARCHITECTURE TRANSFERABILITY

We now focus on black-box settings where the architecture of the target model is unknown (and not used to build the surrogate model). We consider ten architectures (five for both ImageNet and CIFAR-10). Following Liu et al. [2017], Xie et al. [2019], Li et al. [2018], Dong et al. [2018], we hold-out one architecture to act as the target model and use the four remaining ones as surrogates. We apply I-FGSM with 1 model per surrogate architecture per iteration to keep attack cost constant. Due to computational limitations, we limit the training to 135 epochs on ImageNet (3 cycles of 45 epochs for cSGLD). For every architecture, cSGLD and 1 DNN are trained for the same number of epochs.

As shown in Tables 2 (ImageNet) and 3 (CIFAR-10), our method significantly improved transferability on all five hold-out architectures for both datasets, except for the L$\infty$ VGG19 target (with a difference of 0.4 percentage point). On CIFAR-10, the differences range from 15.0 to 35.2 percentage points (2-norm), and from -0.4 to 9.9 ($\infty$-norm). Our method outperforms 4 DNNs per architecture on the L2 attack, despite been trained for 4 times fewer epochs. On ImageNet, cSGLD improves over the one DNN counterpart by 11.8 and 29.9 percentage points of success rate at constant computational train and attack budget.

Appendix E presents the results for an alternative protocol where we use a single architecture as surrogate. In summary, in this setup cSGLD achieves a higher inter-architecture success rate in 39/40 cases on ImageNet, 38/40 cases on CIFAR-10, and in 18/18 cases on MNIST, compared to a single DNN trained for the same number of epochs. Differences range between -0.3 and 44.8 percentage points on ImageNet, -2.3 and 62.1 on CIFAR-10 and 0.2 and 83.2 on MNIST.

We conclude that our method improves transferability even when the target architecture is unknown. This tends to indicate that the adversarial directions against posterior predictive distribution are partially aligned across different architectures. In other words, given a common classification task, the variability of an architecture parameters might be informative of the variability of another architecture parameters.

## 4.3 TEST-TIME TRANSFERABILITY TECHNIQUES

Given that our approach works at train time, we evaluate its combination with test-time techniques. We apply three test-time transformations to cSGLD samples and one DNN obtained with the same number of epochs (300 for CIFAR-10, 135 for ImageNet). The ImageNet surrogates are ResNet50 (respect. PreResNet110 on CIFAR-10). The targets are the same as in Section 4.2. Following Li et al. [2018], Xie et al. [2019], Wu et al. [2020], we also combine every test-time

Table 2: Transfer success rates of I-FGSM attack on ImageNet hold-out architectures. Higher is better.

| Norm | Surrogate | Target Architecture | | | | | Nb epochs |
|---|---|---|---|---|---|---|---|
| | | −ResNet50 | −ResNeXt50 | −DenseNet121 | −MNASNet | −EffNetB0 | |
| L2 | 1 cSGLD per arch. | **93.28** ±0.12 | **90.61** ±0.24 | **92.25** ±0.26 | **95.98** ±0.19 | **81.88** ±0.38 | $4 \times 135$ |
| | 1 DNN per arch. | 72.99 ±0.52 | 72.31 ±0.44 | 64.72 ±0.59 | 84.21 ±0.18 | 53.99 ±0.76 | $4 \times 135$ |
| L∞ | 1 cSGLD per arch. | **92.21** ±0.23 | **89.83** ±0.22 | **90.86** ±0.19 | **95.85** ±0.46 | **79.40** ±0.42 | $4 \times 135$ |
| | 1 DNN per arch. | 69.65 ±0.47 | 69.01 ±0.70 | 61.00 ±0.66 | 82.25 ±0.03 | 49.71 ±1.37 | $4 \times 135$ |

Table 3: Transfer success rates of I-FGSM attack on CIFAR-10 hold-out architectures. The ⋆ symbol indicates that 1 DNN per architecture is better than 1 cSGLD per architecture. Higher is better.

| Norm | Surrogate | Target Architecture | | | | | Nb epochs |
|---|---|---|---|---|---|---|---|
| | | −PResNet110 | −PResNet164 | −VGG16 | −VGG19 | −WideResNet | |
| L2 | 1 cSGLD per arch. | **95.56** ±0.04 | **95.72** ±0.06 | **45.96** ±0.07 | **42.60** ±0.08 | **84.04** ±0.05 | $4 \times 300$ |
| | 1 DNN per arch. | 60.38 ±1.09 | 60.93 ±1.06 | 29.97 ±0.48 | 27.57 ±0.66 | 57.86 ±0.74 | $4 \times 300$ |
| | 4 DNNs per arch. | 77.12 ±1.32 | 77.21 ±1.14 | 40.89 ±0.63 | 40.18 ±0.76 | 77.54 ±0.93 | $4 \times 1200$ |
| L∞ | 1 cSGLD per arch. | 96.38 ±0.06 | 96.51 ±0.08 | 49.19 ±0.06 | 45.17 ±0.03 | 84.75 ±0.01 | $4 \times 300$ |
| | 1 DNN per arch. | 87.02 ±0.04 | 88.86 ±0.04 | 44.99 ±0.10 | ⋆45.55 ±0.02 | 74.84 ±0.03 | $4 \times 300$ |
| | 4 DNNs per arch. | **96.50** ±0.01 | **97.01** ±0.02 | **59.80** ±0.01 | **59.08** ±0.01 | **89.23** ±0.04 | $4 \times 1200$ |

technique with momentum[8].

Table 4 shows the results on ImageNet (Appendix F for CIFAR-10). We observe that our approach and the test-time techniques complement well to each other. Indeed, the best success rates are always achieved by a technique applied on cSGLD (in bold). All three techniques combined with momentum applied on cSGLD achieve a systematically higher success rate than the same technique applied on 1 DNN, with differences ranging from 10.7 to 41.7 percentages points on ImageNet and from 3.8 to 56.2 on CIFAR-10. Overall, the addition of a technique (excluding momentum alone) to our vanilla cSGLD surrogate never decrease the success rate on CIFAR-10 and only in 10% of the averaged cases considered on ImageNet, as indicated by the † symbols.

Besides, our vanilla cSGLD surrogate achieves better transferability than any of the test-time techniques applied to 1 DNN in 90% of the cases on CIFAR-10 and 93.3% on ImageNet, using the I-FGSM attack. Similarly, for MI-FGSM, we observe 76.7% for the former and 90% for the latter. This demonstrates that despite previous efforts in providing effective test-time techniques for transferability (see Section 2), *improving the training of the surrogate – in our case, through efficient sampling from the posterior distribution – yields significantly higher improvements*. Hence, while training approaches have been overlooked, canonical elements that have been related to transferability, ie. skip connec-

tions [Wu et al., 2020], input [Xie et al., 2019] and model diversity [Li et al., 2018], should be put into perspective compared to the importance that the posterior distribution appears to have.

## 4.4 BAYESIAN AND ENSEMBLE TECHNIQUES

In addition to cSGLD and Deep Ensemble, we explore the use of other training techniques to improve transferability: two other Bayesian techniques – Variational Inference (VI) and Stochastic Weight Averaging-Gaussian (SWAG) – and two other ensembling techniques – Snapshot ensembles (SSE) and Fast Geometric Ensembling (FGE). We train each for an equivalent computational cost of 3 DNNs on CIFAR-10 and 2 DNNs on ImageNet (except for VI and SWAG, see discussion in Appendix B). Figure 2 presents the success rate of L∞ I-FS(S)M attack with the corresponding training computational cost (in flops), as we increase the number of models in each surrogate. Appendix B contains details on the methods and the results for L2 I-FS(S)M.

On CIFAR-10, the success rate of the first 4 cycles of cSGLD increases substantially from one cycle to the next (from 76.58% to 81.56% for the first to the second cycle) and within a single cycle (from 81.56% to 87.20% between the start and the end of the second cycle). This reveals that exploring modes of the posterior plays an important role to generate transferable adversarial examples, and that there is some local geometric discrepancy of the loss landscape among local maxima. On ImageNet, transferability

---

[8]All rows with momentum correspond to MI-FGSM, an attack variant designed to improve transferability [Dong et al., 2018].

Table 4: Transfer success rates of (M)I-FGSM improved by our approach combined with test-time techniques on ImageNet (in %). Target in column. ResNet50 is intra-architecture transferability, others are inter-architecture. Bold is best. Symbols ⋆ are DNN-based techniques better than our vanilla cSGLD surrogate, † are techniques that degrades their vanilla surrogate. All techniques improve with cSGLD compared to 1 DNN.

| Norm | Surrogate | Target Architecture | | | | |
|---|---|---|---|---|---|---|
| | | ResNet50 | ResNeXt50 | DenseNet121 | MNASNet | EffNetB0 |
| L2 | 1 DNN | $56.60_{\pm0.71}$ | $41.09_{\pm0.61}$ | $29.73_{\pm0.30}$ | $28.13_{\pm0.17}$ | $16.64_{\pm0.33}$ |
| | + Input Diversity | $83.15_{\pm0.30}$ | $73.17_{\pm0.80}$ | $61.24_{\pm0.58}$ | $58.16_{\pm0.36}$ | $\star42.10_{\pm0.36}$ |
| | + Skip Gradient Method | $65.64_{\pm0.88}$ | $52.75_{\pm0.42}$ | $38.58_{\pm0.55}$ | $43.40_{\pm0.61}$ | $29.11_{\pm0.30}$ |
| | + Ghost Networks | $78.84_{\pm0.46}$ | $62.46_{\pm0.38}$ | $45.76_{\pm0.02}$ | $41.44_{\pm0.58}$ | $25.77_{\pm0.11}$ |
| | + Momentum (MI-FGSM) | $\dagger52.53_{\pm0.80}$ | $\dagger37.15_{\pm0.76}$ | $\dagger26.33_{\pm0.48}$ | $\dagger25.21_{\pm0.42}$ | $\dagger14.74_{\pm0.31}$ |
| | + Input Diversity | $80.81_{\pm0.72}$ | $69.55_{\pm0.83}$ | $56.73_{\pm0.39}$ | $54.16_{\pm0.05}$ | $37.07_{\pm0.03}$ |
| | + Skip Gradient Method | $65.65_{\pm0.95}$ | $53.25_{\pm0.18}$ | $38.79_{\pm0.62}$ | $44.33_{\pm0.63}$ | $29.45_{\pm0.28}$ |
| | + Ghost Networks | $71.50_{\pm0.12}$ | $53.45_{\pm0.65}$ | $37.39_{\pm0.47}$ | $34.53_{\pm0.69}$ | $20.29_{\pm0.36}$ |
| | cSGLD | $84.83_{\pm0.55}$ | $74.73_{\pm0.82}$ | $71.45_{\pm0.56}$ | $60.14_{\pm0.44}$ | $39.71_{\pm0.20}$ |
| | + Input Diversity | **$93.87_{\pm0.19}$** | **$89.12_{\pm0.24}$** | **$88.52_{\pm0.16}$** | **$82.78_{\pm0.28}$** | **$66.13_{\pm0.35}$** |
| | + Skip Gradient Method | $\dagger83.17_{\pm0.85}$ | $\dagger72.79_{\pm1.06}$ | $\dagger66.19_{\pm0.89}$ | $71.71_{\pm0.41}$ | $52.66_{\pm0.31}$ |
| | + Ghost Networks | $92.99_{\pm0.13}$ | $85.69_{\pm0.24}$ | $82.81_{\pm0.42}$ | $72.88_{\pm0.30}$ | $50.30_{\pm0.29}$ |
| | + Momentum (MI-FGSM) | $\dagger82.44_{\pm0.19}$ | $\dagger70.93_{\pm1.04}$ | $\dagger66.19_{\pm0.56}$ | $\dagger55.51_{\pm0.59}$ | $\dagger34.49_{\pm0.59}$ |
| | + Input Diversity | $93.48_{\pm0.23}$ | $87.87_{\pm0.15}$ | $86.81_{\pm0.33}$ | $80.37_{\pm0.20}$ | $60.26_{\pm0.02}$ |
| | + Skip Gradient Method | $\dagger82.35_{\pm0.10}$ | $\dagger71.54_{\pm0.58}$ | $\dagger64.50_{\pm0.18}$ | $70.47_{\pm0.22}$ | $50.80_{\pm0.23}$ |
| | + Ghost Networks | $90.11_{\pm0.18}$ | $80.35_{\pm0.61}$ | $75.10_{\pm0.67}$ | $64.08_{\pm0.12}$ | $39.85_{\pm0.52}$ |
| L∞ | 1 DNN | $47.81_{\pm1.09}$ | $32.29_{\pm0.64}$ | $23.43_{\pm0.32}$ | $22.52_{\pm0.45}$ | $12.77_{\pm0.32}$ |
| | + Input Diversity | $76.55_{\pm1.01}$ | $62.57_{\pm0.56}$ | $50.17_{\pm0.33}$ | $49.31_{\pm0.18}$ | $\star32.64_{\pm0.09}$ |
| | + Skip Gradient Method | $66.36_{\pm0.50}$ | $51.60_{\pm0.36}$ | $39.05_{\pm0.24}$ | $45.60_{\pm0.72}$ | $30.69_{\pm0.03}$ |
| | + Ghost Networks | $67.02_{\pm0.17}$ | $46.74_{\pm0.63}$ | $32.57_{\pm0.17}$ | $31.12_{\pm0.77}$ | $17.68_{\pm0.05}$ |
| | + Momentum (MI-FGSM) | $55.12_{\pm0.82}$ | $38.47_{\pm0.82}$ | $28.19_{\pm0.14}$ | $27.55_{\pm0.67}$ | $16.34_{\pm0.37}$ |
| | + Input Diversity | $\star82.47_{\pm0.41}$ | $\star69.69_{\pm0.81}$ | $57.79_{\pm0.57}$ | $\star55.99_{\pm0.37}$ | $\star38.63_{\pm0.29}$ |
| | + Skip Gradient Method | $68.39_{\pm0.53}$ | $54.57_{\pm0.60}$ | $41.48_{\pm0.37}$ | $47.97_{\pm0.41}$ | $\star33.16_{\pm0.37}$ |
| | + Ghost Networks | $71.27_{\pm0.54}$ | $51.46_{\pm0.84}$ | $36.91_{\pm0.48}$ | $34.54_{\pm0.32}$ | $20.51_{\pm0.30}$ |
| | cSGLD | $78.71_{\pm1.19}$ | $65.11_{\pm1.45}$ | $61.49_{\pm0.59}$ | $51.81_{\pm1.45}$ | $31.11_{\pm0.99}$ |
| | + Input Diversity | $90.03_{\pm0.10}$ | $82.13_{\pm0.45}$ | $81.19_{\pm0.34}$ | $74.48_{\pm0.39}$ | $53.51_{\pm0.39}$ |
| | + Skip Gradient Method | $81.37_{\pm0.72}$ | $69.88_{\pm1.31}$ | $65.20_{\pm0.75}$ | $71.68_{\pm0.53}$ | $52.15_{\pm0.32}$ |
| | + Ghost Networks | $87.33_{\pm0.73}$ | $76.00_{\pm1.33}$ | $71.67_{\pm0.97}$ | $61.45_{\pm0.25}$ | $37.19_{\pm0.68}$ |
| | + Momentum (MI-FGSM) | $82.89_{\pm0.70}$ | $70.42_{\pm1.26}$ | $66.39_{\pm0.74}$ | $56.68_{\pm0.97}$ | $36.00_{\pm1.15}$ |
| | + Input Diversity | **$93.97_{\pm0.26}$** | **$87.69_{\pm0.44}$** | **$86.78_{\pm0.16}$** | **$81.08_{\pm0.14}$** | **$60.87_{\pm0.48}$** |
| | + Skip Gradient Method | $84.19_{\pm0.21}$ | $73.14_{\pm0.99}$ | $67.35_{\pm0.26}$ | $74.36_{\pm0.47}$ | $55.30_{\pm0.16}$ |
| | + Ghost Networks | $89.53_{\pm0.05}$ | $78.69_{\pm0.19}$ | $73.33_{\pm0.58}$ | $63.56_{\pm0.35}$ | $39.79_{\pm0.52}$ |

improves mainly by sampling from several local optima.

Interestingly, even though FGE and SWAG build an ensemble around a single local optimum, their flexibility allows capturing general adversarial directions. The FGE surrogates trained for more than 0.30 petaflops have systematically higher success rates than cSGLD and SSE on CIFAR-10. However, the opposite is observed on ImageNet: FGE is not competitive with methods exploring several local optima (cSGLD, SSE, and Deep Ensemble). We hypothesize that modes are not as well-connected on larger datasets.

The efficiency of SWAG on both datasets opens new directions to create hybrid attacks based on few additional iterations over the training set. SWAG approximates the posterior with a Gaussian fitted on some additional SGD epochs from a pretrained DNN. It captures well the shape of the true posterior [Maddox et al., 2019], reinforcing our views on the strong relationship between the posterior and transferability. The success rate gap between cSGLD/SSE and SWAG on ImageNet suggests higher geometrical discrepancies between local loss maxima on larger datasets.

VI fails to compete with Deep Ensemble on both success rate and computational efficiency for the L∞ attack on CIFAR-10, but beats it on L2 bound and on ImageNet.

On CIFAR-10, the marginal impact beyond 6 cSGLD cycles, 17 SWAG samples, 7 SSE models, and 35 FGE models becomes noisy. We hypothesize that correlated samples pro-

duce these limitations. Hence, the use of multiple runs is a promising direction for greater transferability.

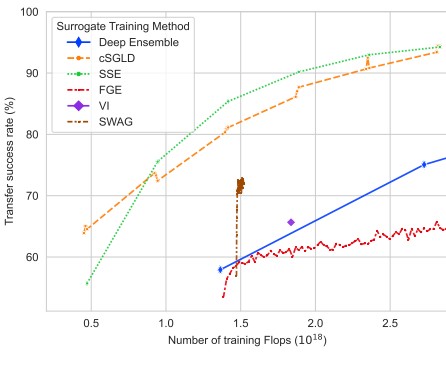

(a) ImageNet

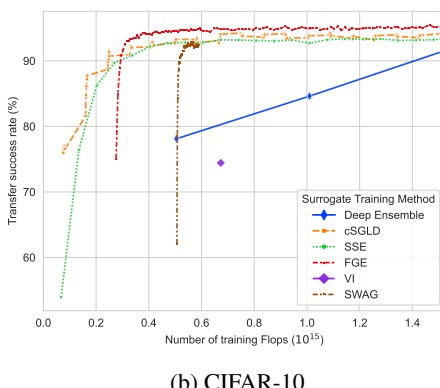

(b) CIFAR-10

Figure 2: Intra-architecture L∞ I-FGSM success rate with respect to the training computational complexity of an increasing number of samples from six training techniques.

### 4.5 THREATS TO VALIDITY

External validity threats arise from the generalization outside the context of the study. First, our results may not generalize to non $p$-norm constrained adversarial examples. However, this way of ensuring imperceptibility is common to all the related work we know on transferability. We also systematically evaluate L2 and L∞ attacks, while most previous studies do not. Second, similar to all competitive approaches, we consider benchmark datasets of classification in computer vision. The generalization of our conclusions to other domains and tasks could require a dedicated study. Finally, we fed adversarial examples directly to the target model. Evaluating adversarial examples through the physical domain may degrade success rates significantly.

Internal validity threats come from the design of the study. Our approach relies on the empirical fact that SGD is approximately a Bayesian sampler [Mingard et al., 2020]. A definitive proof would strengthen the premise of our paper.

Moreover, despite our best effort to control confounding factors, some may exist, such as training hyperparameters.

Threat to construct validity is a consequence of metrics not suitable for evaluation. Our T-DEE metric might not be reliable when the success rate is not increasing with the number of independently trained DNNs. None of our experiences exhibit this (except L2 FSGM on MNIST, see supplementary materials).

## 5 CONCLUSION AND FUTURE WORK

We are the first to extensively investigate training-time approaches to enhance transferability. We discover a strong connection between the posterior predictive distribution and both intra- and inter-architecture transferability. Our Bayesian surrogate is efficient and effective to craft adversarial examples transferable to deterministic DNNs. Our approach further improves existing adversarial attacks and test-time transferability techniques, as one can use it on top of them to perform approximate Bayesian model averaging efficiently and with minimal modifications. We show that our simple training-time approach improves transferability more than previous test-time techniques. We, therefore, cast an important yet overlooked direction to explain transferability and pave the way for new hybrid attacks. Overall, we provide new evidence that the Bayesian framework is a promising direction for research on adversarial examples.

Our studied threat model relates mostly to uncertainty in parameter estimation. A promising venue is to explore how other settings change the types of uncertainty. The ignorance of the training dataset would increase the aleatoric uncertainty. Adding a defence such as random input transformation [Xie et al., 2018] would increase the epistemic uncertainty if its presence is unknown, and the aleatoric uncertainty through its randomness.

Another interesting direction for future work is the transferability to adversarially trained targets. If weight distributions of regular and adversarial training are orthogonal, the latter might be an effective countermeasure to our method.

### Acknowledgements

This work is supported by the Luxembourg National Research Funds (FNR) through CORE project C18/IS/12669767/STELLAR/LeTraon.

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
