# OpenReview forum: "Efficient and Transferable Adversarial Examples from Bayesian Neural Networks"
_auai.org/UAI/2022/Conference — UAI 2022 Poster_

### Official Review · Reviewer_swYF · 2022-04-10

**Q2(1) Originality/Novelty:** 2
**Q2(2) Significance/Impact:** 2
**Q2(3) Correctness/Technical Quality:** 3
**Q2(6) Clarity Of Writing:** 3
**Q6 Overall Score:** 6
**Q8 Confidence In Your Score:** 3

**Q1 Summary And Contributions:**

To improve the transferability of adversarial attacks, this paper introduces a Bayesian deep learning technique to construct the surrogate model, which is trained with a cyclical variant of Stochastic Gradient Markov Chain Monte Carlo (cSGLD) to sample from the posterior distribution of neural network weights. Experiments conducted on CIFAR-10 and ImageNet show the effectiveness of the proposed method.

**Q2 Assessment Of The Paper:**

More detailed information regarding each of these aspects is given below:

**Q2(4) Quality Of Experiments (Optional):**

3: Good: The experimental evaluation is adequate, and the results convincingly support the main claims.

**Q2(5) Reproducibility:**

3: Good: Key resources (e.g., proofs, code, data) are available and key details (e.g., proofs, experimental setup) are sufficiently well-described for competent researchers to confidently reproduce the main results.

**Q3 Main Strengths:**

1) A new idea to understand transferability of adversarial attacks is given by relating the uncertainty and transferability with a Bayesian perspective.

2) A new method to generate transferable adversarial examples is proposed by introducing a Bayesian deep learning technique to approximately sampling neural network weights and perform approximate Bayesian model averaging.

3) This method not only makes the Bayesian surrogates more efficient and effective to craft adversarial examples transferable to deterministic DNNs, but also further improves existing adversarial attacks and test-time transferability techniques.

4) A new metric (T-DEE) is provided to compare the effectiveness of transferability techniques.

5) Extensive experiments ware conducted to verify the effectiveness of the method.


**Q4 Main Weakness:**

1) The novelty of this paper is unclear and should be further improved, since it is trivial to generate transferable adversarial examples by introducing an existing Bayesian deep learning model (i.e., cSGLD).

2) This paper concludes that the posterior predictive approximates an optimal surrogate model under some conditions, but more theoretical interpretations or proofs should be provided.

3) The relationship among posterior distribution of weights, target prior, intra/ inter -architecture, and known/unknown architecture of the target model should be clearly explained.



**Q5 Detailed Comments To The Authors:**

1) What’s the technical contribution of generating transferable adversarial examples? I suggest the authors discuss the necessity and rationality of SG-MCMC and cSGLD in the introduction.

2) This paper concludes that the posterior predictive approximates an optimal surrogate model under some conditions. Thus, more theoretical interpretations or proofs should be given.

3) What is the relationship among posterior distribution of weights, target prior, intra/inter-architecture, and known/unknown architecture of the target model?




**Q7 Justification For Your Score:**

This paper proposes a Bayesian deep learning technique to improve the transferability of adversarial attacks. Extensive experiments show the effectiveness of the proposed method. Thus, I suggest weak accept.

**Q9 Complying With Reviewing Instructions:**

1: Yes.

---

### Official Review · Reviewer_9DVR · 2022-04-12

**Q2(1) Originality/Novelty:** 2
**Q2(2) Significance/Impact:** 2
**Q2(3) Correctness/Technical Quality:** 3
**Q2(6) Clarity Of Writing:** 2
**Q6 Overall Score:** 6
**Q8 Confidence In Your Score:** 1

**Q1 Summary And Contributions:**

The paper presents several contributions, maybe too much regarding the lenght of the paper, conducting the author to add more than 10 pages of supplementary materials.


**Q2 Assessment Of The Paper:**

More detailed information regarding each of these aspects is given below:

**Q2(5) Reproducibility:**

2: Fair: Key resources (e.g., proofs, code, data) are unavailable but key details (e.g., proof sketches, experimental setup) are sufficiently well-described for an expert to confidently reproduce the main results.

**Q3 Main Strengths:**

The paper presents several contributions, maybe too much regarding the lenght of the paper.

**Q4 Main Weakness:**

The paper presents several contributions, maybe too much regarding the lenght of the paper, conducting the author to add more than 10 pages of supplementary materials.


**Q5 Detailed Comments To The Authors:**

I suggest to limit the paper to the main contribution, and publish the whole set of contributions in a (long) journal paper, allowing a more pedagogical approach (true introduction, definitions of all required concepts, notations. )


**Q7 Justification For Your Score:**

I am really sorry to say that I can not review that paper. I am not familiar with Deep Neural Network, neither with adversarial attacks nor with question of transferability. Moreover, since the paper is very dense, key concepts and notations are not all defined, making the review really difficult for a non expert reader.


**Q9 Complying With Reviewing Instructions:**

1: Yes.

---

### Official Review · Reviewer_xZK1 · 2022-04-13

**Q2(1) Originality/Novelty:** 3
**Q2(2) Significance/Impact:** 2
**Q2(3) Correctness/Technical Quality:** 3
**Q2(6) Clarity Of Writing:** 3
**Q6 Overall Score:** 5
**Q8 Confidence In Your Score:** 2

**Q1 Summary And Contributions:**

The paper proposes a novel and efficient method to improve the transferability of adversarial examples using approximate Bayesian inference for building surrogate models. The paper also presents a metric (T-DEE) for approximating the transfer effect.

**Q2 Assessment Of The Paper:**

More detailed information regarding each of these aspects is given below:

**Q2(4) Quality Of Experiments (Optional):**

3: Good: The experimental evaluation is adequate, and the results convincingly support the main claims.

**Q2(5) Reproducibility:**

3: Good: Key resources (e.g., proofs, code, data) are available and key details (e.g., proofs, experimental setup) are sufficiently well-described for competent researchers to confidently reproduce the main results.

**Q3 Main Strengths:**

Novel and efficient approach with improved transfer rates on four state-of-the-art attack methods. A novel metric of for approximate transfer effect.

**Q4 Main Weakness:**

The perspective of countermeasures on the proposed attack approach is not addressed. Further, an extended discussion on the validity threats related to the experiments would also have added value to the paper, e.g. internal and external validity threats.

**Q5 Detailed Comments To The Authors:**

Nicely written and presented paper, which identifies a relevant gap in existing research. The proposed approach seems fair and shows interesting results on the two included datasets. A discussion on the validity threats to the experiments would have increased the transparency of the study. Such a discussion could also be extended to address the countermeasure perspective related to the proposed surrogate attack method. The paper is nicely written and presented but it could have been more clear in the text which tables/figures are included in the paper, and which are placed in the appendix e.g., Table 6 and Figure 7.

**Q7 Justification For Your Score:**

My overall score is based on a general analysis of the pros and cons listed in Q1-Q5 above.

**Q9 Complying With Reviewing Instructions:**

1: Yes.

---

### Official Review · Reviewer_A3iY · 2022-04-18

**Q2(1) Originality/Novelty:** 2
**Q2(2) Significance/Impact:** 2
**Q2(3) Correctness/Technical Quality:** 2
**Q2(6) Clarity Of Writing:** 2
**Q6 Overall Score:** 5
**Q8 Confidence In Your Score:** 3

**Q1 Summary And Contributions:**

The paper improves the transferability of adversarial examples by studying the surrogate model from a Bayesian perspective.

**Q2 Assessment Of The Paper:**

More detailed information regarding each of these aspects is given below:

**Q2(4) Quality Of Experiments (Optional):**

2: Fair: The experimental evaluation is weak: important baselines are missing, or the results do not adequately support the main claims.

**Q2(5) Reproducibility:**

2: Fair: Key resources (e.g., proofs, code, data) are unavailable but key details (e.g., proof sketches, experimental setup) are sufficiently well-described for an expert to confidently reproduce the main results.

**Q3 Main Strengths:**

- Technique: The paper presents a novel technique to generate more transferable adversarial examples with Bayesian deep learning.

- Presentation: I find the paper is well written and easy to follow.

- General result: The result shows the proposed method is compatible with existing techniques and significantly improve adversarial examples' transferability.

**Q4 Main Weakness:**

- Comparison with other SG-MCMC methods: The paper proposes to use cSGLD without stating why it is the best choice.

- Methodology novelty: The paper borrows the existing technique, mainly cSGLD. The claimed technical contribution, T-DEE, is a simple adaptation from the existing metric DEE.




**Q5 Detailed Comments To The Authors:**

Besides the weakness mentioned in Q4. I would like to discuss with the author the explanation of why [1] and this paper have different conclusions about BNN's gradient. Basically, the paper [1] claims BNN is harder to attack and shows adversarial robustness. The author claims attacking BNN is viable and it even leads to more transferable attacks. Thus there is a contradiction. The author provides the following explanation.

1. "exploit the cyclical nature of cSGLD to avoid fully vanished gradients". I am not sure I understand this point. Could the author further explain it?

2. "our cheaper SG-MCMC, ..., may produce correlated MCMC samples, which explains that the averaged gradient is not zero". Generating correlated MCMC samples is usually considered harmful to the sampling process and slows down the whole procedure. Is the author, here, saying the sampling method used here is not very good?

3. "we consider larger datasets (ImageNet and CIFAR-10) than them (MNIST and Fashion-MNIST), where convergence to flat loss is harder to reach." Does it mean the proposed method will fail on small datasets, like MNIST and Fashin-MNIST, since the author seems to admit that the vanishing gradient issue does exist when the dataset is small?

As the author can see, as a reader, I am not satisfied with the explanation in the current paper. I suggest the author investigate further and find deeper reasons. I would like to see, the performance of using the sampling method in [1], does it also generates transferable adversarial examples? In addition, I would like to see the performance on MNIST and Fashion-MNIST. Does the proposed method still work in small datasets? It would be awkward if there exists a threshold on the amount of data for the proposed method to be effective.

[1]  Robustness of Bayesian neural networks to gradient-based attacks

**Q7 Justification For Your Score:**

Overall, I like this work. This paper has a new idea and convincing results. However, I do have issues explained above that hinder me from giving a high score. I hope the author finds my comments helpful. I will change my score if the author addresses my concerns.

**Q9 Complying With Reviewing Instructions:**

1: Yes.

---

### Decision · Program_Chairs · 2022-05-15

**Decision:**

Accept (Poster)

**Comment:**

Meta Review: All 4 reviewers see value in this submission and all 4 favour acceptance (albeit not strongly). The authors have carefully considered critical feedback from reviewers. These are solid reasons for acceptance.